# The Workaholism–Technostress Interplay: Initial Evidence on Their Mutual Relationship

**DOI:** 10.3390/bs13070599

**Published:** 2023-07-17

**Authors:** Carmela Buono, Maria Luisa Farnese, Paola Spagnoli

**Affiliations:** 1Department of Psychology, University of Campania “Luigi Vanvitelli”, 81100 Caserta, Italy; paola.spagnoli@unicampania.it; 2Department of Psychology, Sapienza University of Rome, 00185 Roma, Italy; marialuisa.farnese@uniroma1.it

**Keywords:** technostress, workaholism, cross-lagged panel study, COVID-19, work conditions

## Abstract

During the pandemic, the occurrence of extreme working conditions (e.g., the sudden shift to remote work, isolation, and the slowdown of the work processes) exacerbated several phenomena, such as increased workaholism and stress due to technological devices; that is, technostress. Literature on the onset of these phenomena during the pandemic highlighted a possible interplay among them; however, there is still a dearth of knowledge about the direction of the relationship between workaholism and technostress. The present study assessed the relationship between workaholism and technostress through a two-wave cross-lagged study using path analysis in SEM (Structural Equation Modeling). The study was conducted in Italy during the pandemic, and a total of 113 Italian employees completed the online survey at each wave. Results showed that workaholism at Time 1 was a significant predictor of technostress at Time 2 (β = 0.25, *p* = 0.049), while the reversed causation was not supported (β = 0.08, *p* = 0.22). These findings may help employees and organizations to better understand the phenomena of technostress and workaholism and develop strategies to prevent the consequences of excessive and compulsive work and to improve the balanced use of technology for their daily activities.

## 1. Introduction

Nowadays, the use of information and communication technologies (hereafter, ICTs) is an essential part of many jobs, and their importance has been further emphasized during the COVID-19 pandemic [1] due to the widespread remote work that led to an acceleration of digital transformation [2]. As a result, the digital transformation and the increased reliance on ICTs have had a significant impact on work dynamics, leading to a shift from traditional office work to remote work [3], greater work extension [4], and work intensification [5,6,7]. Work extension is associated with ICT-based organizational practices which blur boundaries between work and non-work time, for instance, performing work commitments during non-work time and space [8], thus not allowing workers to detach themselves from work. Work intensification is associated with organizational practices that encourage employees to work harder and exert more effort, being facilitated by the use of ICTs [4].

Scholars showed that work-related ICT use allows obtaining several benefits to employees’ well-being, such as increased flexibility and a sense of autonomy, which in turn improve their ability to better manage the interface between work and personal life [9,10]. However, the use of ICTs may also pave the way to negative consequences, allowing the employees to stay connected to work without temporal or spatial boundaries, including the invasion of work-related ICT use into the sphere of private life [11], to handle ever-increasing workloads [12]. In this context, supervisors show high expectations of employees’ availability which, in turn, may lead employees to perceive themselves as being expected to be available outside of their working hours [13,14]. In addition, Kondrysova and colleagues [15] showed that employees who perceive that they are required to be available outside of work hours more often resort to using a smartphone and experience less psychological detachment [16,17]. In turn, availability expectations appear to be positively associated with negative outcomes such as problems with sleep and family situations [18].

The temporal and spatial intensive use of ICTs, that is, the possibility to work anywhere and anytime, often hinders psychological recovery [19,20] and leads employees to technostress, a specific type of stress due to the use of ICT arising from the inability to adapt or healthily cope with them [21]. A context of work intensification and work extension may also pave the way to the risk of developing workaholism, which is an intense need for work [22] characterized by strong absorption in work experience and difficulty in disconnecting from job requirements [23]. In this perspective, workaholism and technostress could be associated. On the one hand, working compulsively and excessively can lead to maladaptive use of technology [24]; thus, employees with workaholism tendencies might use technology too intensively (resulting in potential technostress) because of their strong compulsion to work. On the other hand, technology that allows for extended normal working hours could bring employees’ compulsive need to work more and more incessantly [25]. This interplay can lead to a resource loss cycle that may negatively affect employees’ well-being.

Although workaholism and technostress often result intertwined, very low attention was addressed to the study of the relationship between these constructs, leaving unexplored the issue related to their temporal sequence. An exception is a study by Spagnoli and colleagues [26], highlighting that employees with high levels of workaholism show higher levels of technostress when the level of authoritarian leadership is high. We believe that a deeper understanding of this interplay may allow researchers and practitioners to better investigate and improve the quality of working life, preventing negative phenomena such as workaholism and technostress. In view of these hints, the aim of the current study is to fill this research gap by exploring whether technostress due to ICT intensive use leads to employees’ workaholism or vice versa.

The first section of this study provides a theoretical overview of technostress, workaholism, and the relationship between these two constructs. In the second major section, the study describes the materials and methods used (participants, measures, statistical analysis). In the third section, we report the results of the study. Finally, the last section discusses the results in light of the original contribution provided, as well as its limitations.

## 2. The Relationship between Technostress and Workaholism

According to the World Health Organization [27], the increased use of ICTs has modified work patterns and raised organizations’ expectations about the constant availability of employees [28]. Indeed, although the intensive use of technology—both remotely and in the workplace—has facilitated and speeded up some processes (e.g., data processing), it also implies new job design features and demands that can pave the way to the so-called technostress [25], “the phenomenon of stress experienced by end users in organizations as a result of their use of ICTs” (Ragu–Nathan et al. [29] (pp. 417–418). Tarafdar and colleagues [30] highlighted possible processes in the use of ICTs that may enhance the perceived stress, defining technostress as “the stress that users experience because of application multitasking, constant connectivity, information overload, frequent system upgrades, and consequent uncertainty, continual relearning and consequent job-related insecurities, and technical problems associated with the organizational use of ICT” [30] (pp. 304–305). Drawing on Brod’s [21] definition, Ragu–Nathan and colleagues [29] identified five main technostress factors which may cause this specific type of stress: the techno-overload, where ICTs increase the volume of work and induce employees to work faster and longer; the techno-invasion, where ICTs overwhelm employees’ personal life, making them feel to be reached anywhere and anytime; techno-complexity, where ICTs are subject to continuous updates, boosting the employees’ need to revise their knowledge continuously; the techno-insecurity, where employees fear losing their job and being overcome by others that manage ICTs better; and the techno-uncertainty, where continuous updates in ICTs make employees unconfident about their own knowledge. Molino and colleagues [25] suggest that the first three dimensions (namely, techno-overload, techno-invasion, and techno-complexity) are the most relevant in today’s scenario. In line with them, the study of Rohwer and colleagues [31] showed that the techno-invasion and techno-overload dimensions were the most frequently examined in the literature. Literature suggests that these factors the ICT underpins may, in turn, cause technostress [32]. Consistent with the JD-R model, research showed that when work demands are extremely high, extra effort is required to achieve task goals, and this, in turn, may reduce employee resources due to a higher likelihood of developing physical and psychological consequences, such as exhaustion, irritability [33] and higher levels of stress [34]. The extensive adoption of ICTs has also encouraged a climate of work overload, inducing employees, particularly those workaholics, not to stop working and to handle ever-increasing workloads [23,35]. Workaholism includes attitudinal and behavioral components, being defined as “a multidimensional construct composed of (1) an inner pressure or compulsion to work (i.e., motivational dimension); (2) persistent, uncontrollable thoughts about work (i.e., cognitive dimension); (3) feeling negative emotions when not working or when prevented from working (i.e., emotional dimension), and (4) excessive working that goes beyond what is required and expected (i.e., behavioral dimension)” [36] (p. 7). Hence, the general attitude toward work and related behaviors makes workaholism a strong predictor for overtime for these employees, who spend a great deal of their time and energy in their job far beyond what is due [37]. The JD-R model suggests that when workaholics spend excessive amounts of energy and effort at work or in extra-work time, also thanks to ICT, they might exhaust their resources coming to experience stress [38]. With these points in mind, it is reasonable to expect that overuse of ICTs may, among others, affect the employees’ attitude toward their work, inducing workaholism as a dysfunctional coping response. However, extant research suggests that workaholism is an important precursor of work intensification and extension; hence, we can also expect that it may affect an increased use of ICTs, acknowledged as a tool to support their compulsion to work.

Over the past decades, the literature’s attention has been focused on the role of individual characteristics in the adoption of workaholic behaviors. A status perspective underlines how the employees’ heavy work investment may be informed by situational factors, such as the organizational climate, culture, or process design [39,40,41]. In overwork contexts (e.g., with a highly performance-oriented climate or heavy levels of job demands), technology makes it possible to extend normal working hours and perform supplemental job tasks during non-work hours, resulting in higher effectiveness, but also in potential technostress. According to this perspective, the work context may foster the insurgence of workaholism [42,43] when high levels of technostress solicit the employees’ compulsive need to work harder and incessantly [25] to be aligned with the organizational demands and to cope with the negative psychological state associated with the use of ICTs. Hence, techno-stressors could act as job demands, the resulting workaholic behavior representing the dysfunctional coping strategy to respond to them.

On the other hand, workaholism may be conceived as the expression of a stable personal characteristic, a trait-like phenomenon [44], or a state condition with within-individual variation [45]. Drawing on self-determination theory (SDT) [46], we can suppose that in this case, employees high in trait workaholism should be motivated by a strong and uncontrollable need to work incessantly (controlled motivation) feeling aligned with high job demands; this can lead them to devote more and more time and resources to work. These workaholics’ impulses have become easy to satisfy due to advances in technology [47,48], and in this light, technology becomes a resource that workaholic employees tend to use more frequently (compared to non-workaholic ones), allowing them to be more effective (e.g., timely checking their work emails or talking to their supervisor anytime, anywhere). Although state workaholism may be experienced by both workaholic and non-workaholic individuals, it is reasonable to expect that employees with workaholic traits are more likely to respond to work-related demands by exerting higher levels of effort at work. For instance, they may engage in multitasking by performing two or three different activities simultaneously or willingly accepting additional job-related tasks. These behaviors are often observed in individuals with workaholic traits who demonstrate a strong drive to meet work demands, even to the point of exceeding what is typically expected. Indeed, one of the tenets of the whole trait theory of personality [49] is that those who are higher on a trait enact more frequently the related states in daily situations. Overall, intensive use of ICTs can, in turn, lead to perceived technostress in several ways. For instance, ICTs increase the volume of work and induce employees to work faster and longer, thus affecting their perception of techno-overload; when ICTs overwhelm employees’ personal life, making them feel to be reached anywhere and anytime, this may enhance their perception of techno-invasion; also, the ICTs continuous need of updates boosters the employees’ need to update their knowledge, leading to a higher perception of techno-complexity [25,29]. In this perspective, workaholism could be a predictor of technostress. 

To our best knowledge, few studies empirically investigated the technostress–workaholism relationship, overall suggesting a positive relationship, although providing support for both the supposed directions (workaholism affecting technostress or, conversely, technostress affecting workaholism). For instance, Molino and colleagues [25] found an increased risk of technostress in smart workers who had a high workload: particularly, when employees felt that they had to work faster and longer, they also perceived a higher invasion of technology into their private life (namely, a techno-invasion). Other scholars focused on the phenomenon of techno-addiction, defined as “a specific technostress experience due to an uncontrollable compulsion to use ICT everywhere and anytime and to use it for long periods of time in an excessive way” [50] (p. 424) showing as compulsive Internet use at Time 1 was related to compulsively work at Time 2 [51]. Conversely, other studies showed that workaholism led to intensive smartphone use [52] and affected the occurrence of technostress when moderated by an authoritarian leadership style [26]. On the whole, this initial empirical evidence does not allow a clear understanding of the nature of the technostress–workaholism relationship, letting how they relate to each other. The current study aims to fill this research gap by exploring this relationship between them longitudinally and by testing and comparing both possible relationships:

**Hypothesis 1 (H1):** 
*Workaholism at Time 1 will predict a relative increase in technostress at T2.*


**Hypothesis 2 (H2):** 
*Technostress at Time 1 will predict a relative increase in workaholism at T2.*


## 3. Materials and Methods

### 3.1. Participants and Procedure

In line with the purpose of the research, the current study adopted a two-wave longitudinal panel design, collecting data on the same two variables at T1 and T2. The current study was conducted in Italy, using a convenience sampling method. At both times, data were collected through an online questionnaire. Graduate students completing a course in Work and Organizational Psychology voluntarily assisted with data collection. They were asked to contact a limited number of available workers to participate in the study by sending them the link to the online questionnaire to be completed twice: in April 2021 (T1) and three months later (July 2021, T2), i.e., at the peak of the third wave of SARS-CoV-2 in Italy, when all organizations had to reshape their processes by introducing remote working practices and requiring their workers to use ICT extensively. A total of 718 participants completed the survey at T1. After three months, a total of 113 Italian workers also completed the second questionnaire. The T1 and T2 questionnaires were matched using the email provided by the respondents. This procedure ensured that responses were anonymous while allowing the linking of baseline (T1) and follow-up (T2) data. All participants were informed via email about the research objectives, and they signed informed consent about the confidentiality and anonymity aspect of the data. The procedure was conducted in line with the Helsinki Declaration [53] as well as the data protection regulation of Italy (Legislative Decree No. 196/2003) and the European General Data Protection Regulation (GDPR, 2016/679). 

The G*Power analysis method was used to estimate the sample size [54]. Specifically, the sample size was calculated according to the medium effect size (f 2 = 0.25) [55] and a 0.05 α level. G*Power analysis results suggested that at least 106 participants were needed to achieve a statistical efficiency of 0.80. Our study was conducted on 113 Italian employees, which was a sufficient sample size. Finally, in the current study, only participants who were employed at both baseline T1 and T2 have been included. 

### 3.2. Measures

Workaholism was measured with the Multidimensional Workaholism Scale (MWS) [36]. The MWS captures the respondent’s feelings about his/her work, reflecting four components of workaholism: the motivational component (example item “I work because there is a part inside of me that feels compelled to work”); the cognitive component (example item “It is difficult for me to stop thinking about work when I stop working”); the emotional component (example item “I am almost always frustrated when I am not able to work”); and the behavioral component (example item “I tend to work longer hours than most of my coworkers”). These components are each measured with four items, with a 5-point Likert scale from 1 = strongly disagree to 5 = strongly agree. Reliability analysis showed acceptable values. Cronbach alpha for the overall MWS score was 0.88 at T1 and 0.93 at T2 (specifically, Cronbach alphas were T1 = 0.65 and T2 = 0.85 for the motivational component; T1 = 0.90 and T2 = 0.93 for the cognitive component; T1 = 0.84 and T2 = 0.92 for the emotional component; and T1 = 0.86 and T2 = 0.85 for the behavioral component). 

Technostress was measured with the Technostress creators scale [29] through the Italian version adapted by Molino and colleagues [25]. Technostress creators were assessed by 11 items related to the techno-overload factor (example item “I am forced by technology to work much faster”; 4 items), the techno-invasion factor (example item “I spend less time with my family due to technology”; 3 items), and the techno-complexity factor (example item “I do not know enough about technology to handle my job satisfactorily”; 4 items). Participants used a Likert scale from 1 = strongly disagree to 5 = strongly agree. Reliability analysis showed acceptable values. Cronbach alpha for the overall Technostress creators score was 0.89 at T1 and 0.91 at T2 (specifically, Cronbach alphas were T1 = 0.90 and T2 = 0.92 for techno-overload, T1 = 0.78 and T2 = 0.84 for techno-invasion, and T1 = 0.90 and T2 = 0.90 for techno-complexity).

### 3.3. Data Analysis

Analyses were conducted with IBM SPSS Version 21 and AMOS Version 22. First, the SPSS 21 statistical program (IBM) [56] was used to perform descriptive statistics, zero-order correlations, and Cronbach’s alpha coefficients. Zero-order correlations were used to examine the associations between variables, and reliability analysis (Cronbach’s alpha) was used to assess the internal consistency of the scale. Second, the measurement model was tested in AMOS. Third, to explore the proposed model, a two-wave cross-lagged design was carried out by using AMOS. Both the workaholism and technostress factors were specified as latent constructs. For this purpose, item parcels were computed and used to measure latent variables [57]. Items were parceled following subdimensions as indicators: parcels are measures constructed by summing items within a subscale. As indices of the model fit, the following fit indices were considered: CFI (Comparative fit index); RMSEA (Root mean square error of approximation); χ2 (chi-square test); Tucker–Lewis index (TLI) e SRMR (Standardized root mean square residual). CFI assesses the extent to which the tested model is superior to an alternative model in reproducing the observed covariance matrix [58]. The RMSEA introduces a correction for lack of parsimony [59]. The SRMR is an index of the average of standardized residuals between the observed and the hypothesized covariance matrices [60]. Values higher than 0.90 for CFI and TLI and lower than 0.08 for SRMR and RMSEA indicate an acceptable fit to the data.

## 4. Results

### 4.1. Characteristics of Participants

Most of them were female (63.7%), with ages ranging from 23 to 67 years (M = 38.69, SD = 12.97). Education was distributed as follows: the majority of participants had a Bachelor’s or Master’s degree (77%), and the remaining had a high school (22.1%) and middle school (0.9%) diploma. The participants were employed in the private (55.2%) and public (47.8%) sectors. They were employed in management positions (3.5%), freelancers (6.2%), clerks (78.8%), and temporary workers (11.5%). Most of them (45.1%) worked in a hybrid model (both remotely and in presence), 34.5% worked remotely, and finally, 20.4% worked in presence.

### 4.2. Items Analysis

The first step of the analysis was conducted to examine descriptive statistics and normality of the items’ distributions for the Multidimensional Workaholism Scale and Technostress creators. Table A1 in Appendix A shows the standard deviation, MD, skewness, and kurtosis for all items. Skewness and kurtosis values were distributed between −2.0 and +2.0, indicating univariate normality [61,62]. In addition, multivariate normality was tested. No multivariate outliers were detected based on Mahalanobis distance computations.

### 4.3. Attrition Analysis

To test whether those who dropped out of the study (i.e., did not complete the T2 survey) differed from those who stayed, we performed an ANOVA analysis comparing the two subsamples’ means at T1 for each of the study’s variables (workaholism and technostress). The test showed no significant differences, indicating no selection bias for the stayers on these variables. 

### 4.4. Bivariate Correlations

Table 1 shows Pearson correlation, means, standard deviations, and Cronbach alpha coefficients for all variables. The results of bivariate correlations show that workaholism is positively related to the Technostress creators’ score across all two-time points (’s range: 0.26 to 0.42).

### 4.5. Test of Measurement Model

Prior to testing the proposed cross-lagged structural models, dimensionality was tested with confirmatory factor analysis (CFA). The factorial structure of workaholism and technostress scales was assessed using Amos 22 with maximum likelihood estimation [63]. We assessed the fit of the model for technostress and workaholism scales for each time point separately. As shown in Table 2, all model fit values were adequate for both time points (CFI and TLI > 0.90, RMSEA and SRMR 0.08), whereas RMSEA for technostress at Time 2 seemed to be unsatisfactory. However, Kenny and colleagues [64] pointed out that with small degrees of freedom, the RMSEA too often falsely indicates a poorly fitting model. In general, it seems that with samples <500, RMSEA might incorrectly suggest that models do not fit closely.

### 4.6. Analysis of the Cross-Lagged Paths

To investigate the reciprocal relationships between workaholism and technostress was used AMOS with maximum likelihood estimation (see Figure 1). The first model (Model 1) tested was an autoregressive model (no lagged effects), which assumed that the only predictors of the variables at T2 were the same variables at T1. The second model (Model 2) added a cross-lagged pathway from workaholism at T1 to technostress at T2. The third model (Model 3) tested the reverse effect, adding a pathway to the autoregressive model from technostress at T1 to workaholism at T2. The last model (Model 4) tested a reciprocal effect that allowed for the estimation of both cross-lagged effects simultaneously. Table 3 reports summary fit statistics for all analyses, and Table 4 reports the stability and cross-lagged path coefficients and critical ratios. Figure 2 shows the models tested. A critical ratio that approaches or exceeds an approximate absolute value of |1.96| is indicative of a salient parameter estimate [65]. Model 2 (χ^2^_(65)_ = 82.55, *p* = 0.070, CFI = 0.98, TLI = 0.97, RMSEA = 0.049, SRMR = 0.05), testing the cross-lagged pathway from workaholism T1 to technostress T2, was significant (β = 0.26, CR = 8.16, *p* = 0.04). Model 3 (χ^2^_(65)_ = 85.13, *p* = 0.048, CFI = 0.97, TLI = 0.96, RMSEA = 0.053, SRMR = 0.06), incorporating the reverse cross-lagged pathway from technostress at T1 to workaholism at T2, was not significant (β = 0.09, CR = 6.90, *p* = 0.18). Lastly, Model 4, which included both cross-legged pathways, showed an acceptable fit (χ^2^_(64)_ = 81.09, *p* = 0.073, CFI = 0.98, TLI = 0.97, RMSEA = 0.049, SRMR = 0.06), and a significant effect from workaholism at T1 to technostress at T2 (β = 0.25, CR = 1.96, *p* = 0.049) while a not significant effect from technostress at T1 to workaholism at T2 (β = 0.08, CR = 1.21, *p* = 0.22). Finally, the regression of workaholism at Time 1 on technostress at Time 2 has a critical ratio (CR) value of 1.96, thus approaching statistical significance; while it did not exceed this standard for the converse relationship between technostress at Time 1 and workaholism at Time 2.

## 5. Discussion

The present study aimed to investigate the relationship between workaholism and technostress over time, to understand whether workaholism was a possible antecedent of technostress or vice versa. Results indicated that workaholism affected the perceived technostress, confirming the first hypothesis of the current study (H1), while the reverse relationship (H2) was not confirmed. 

This finding is consistent with the trait perspective of workaholism and the literature on its negative health-related consequences [66]. Indeed, in line with the self-determination theory [46], workaholics are motivated by a strong and uncontrollable work motivation, which, in turn, could lead them to make intensive use of ICTs as enablers to stay connected with work, to create new work demands for themselves constantly, and to work beyond what is reasonably expected [67]. In other words, this increase in technology use may help them to better cope with the work demands. Conversely, our findings did not confirm the second hypothesis (H2), showing any significant effect from technostress to workaholism, hence suggesting that technostress may be mainly an outcome rather than a predictor of workaholism. Indeed, individuals higher in workaholism seem to respond differently to these demands, increasing their own risk of technostress. These results are consistent with the literature on technostress that identifies, as possible technostress antecedents, the individuals’ attitudes (e.g., dispositions to the job) [68], personality [69], and cultural values [70]. Overall, our findings support a trait perspective, highlighting a prominent role of workaholism in affecting perceived technostress and showing as this process does not occur in a vacuum; rather, it seems to be enacted by the attitudes and motivations of individuals, where their different degree of workaholism leads them to interpret the demands of the context (e.g., role expectations, workload, performance standards) as affordable challenges and ICT as potential resources.

### 5.1. Practical Implications

The findings of this study also provide some suggestions for monitoring the risk of workaholism and any signs of technostress. There are three levels of prevention of workaholism, and the primary level of prevention involves the work environment; indeed, employees with a workaholic attitude tend to respond to contextual demands with an even greater effort at work. To avoid the consequences of workaholism, such as technostress, it is important that individuals learn to dedicate time to recovery [71]. Bearing in mind that the work context is a crucial variable to be taken into consideration in the insurgence of workaholism, it would be appropriate for organizations to design healthy work environments and promote work behaviors, policies (e.g., rewards), and a consistent value system discouraging workaholic attitudes. Based on reports from European Agency for Safety and Health at Work [72], 46% of respondents said they are exposed to severe time pressure or work overload. Organizations need to foster a culture in which recovery is considered an important factor for workplace well-being [45]. Balducci and colleagues [45] argued that a lack of recovery due to an exclusive investment in work-related activities could lead to distress in workaholics. Furthermore, Huyghebaert and colleagues [73] and Falco and colleagues [74] proposed that organizations can prevent workaholism by encouraging work–life balance through well-defined organizational segmentation and specific work schedules. It is important to acknowledge that the increased use of ICT for flexible work has resulted in blurred boundaries between work and private life, as individuals remain constantly connected.

Furthermore, the organizational policies should monitor the workload levels and reduce or avoid supplemental work supported by technological tools. Andreassen and colleagues [75] showed that reducing job demands may have positive effects in terms of both workaholism and health. Hassler and colleagues [76] showed that workers who were constantly connected claimed they had financial advantages, and this was beneficial to their career progression. Organizations should, therefore, take into consideration these potential risks induced by ICTs, especially for workaholics. In addition, supervisors, representing an influential model for work behaviors, should be good examples of working in a healthy way [77]. Finally, organizational support and training can assist employees in effectively managing ICT. This support may include, for instance, measures such as reducing the perceived complexity of ICTs or addressing employees’ concerns about job insecurity and the potential risk of losing their jobs.

Overall, in light of our findings, designing processes and organizational culture that are recovery-oriented and respectful of work–life balance would be beneficial to workers not only because of the direct effect of these policies but also because it would help workaholics interpret the context—and thus also the relationship with technologies—providing them a framework that would discourage a dysfunctional and stressful use of ICT.

The secondary level of prevention involves identifying workaholic behavior by observing specific work habits such as spending excessive hours at the office or engaging in late-night work-related emails [78]. Health professionals can use these observable indicators as cues to identify individuals who may be exhibiting workaholic tendencies [78]. Although organizations play a crucial role in preventing workaholism, it is also essential to act at the individual level. For instance, counseling based on self-validation [79] can help workaholics to disengage from work and engage in extra-work activities. Additionally, several studies showed that mindfulness meditation could be used to prevent workaholism and to improve work-related well-being [79].

Mindfulness-based interventions have been found to help employees develop skills for self-regulating their emotions [80]. Future studies could explore the effects of training interventions that target specific components of technostress, for example, improving digital skills to reduce the perceived complexity of using ICTs or alleviating uncertainty about staying updated with technological advancements. These interventions could specifically focus on workaholic employees, aiming to enhance their confidence and ability to regulate the pace of technology use. Hence, these studies could provide insights into effectively supporting workaholic individuals in managing their relationship with technology.

The tertiary level of prevention focuses on rehabilitation and aims to support workaholics in recovering from the negative consequences of their behavior. This level of prevention recognizes that some individuals may already be experiencing adverse effects due to workaholism and seeks to provide assistance and interventions to facilitate their recovery. The rehabilitation process may involve various strategies such as counseling, therapy, support groups, and implementing changes in work habits and lifestyle to promote a healthier work–life balance. For instance, Falco and colleagues [74] suggest the use of cognitive behavioral interventions to help workaholics reduce maladaptive emotions and irrational cognitions, which underlie workaholism. Balducci and colleagues [45] suggest self-help groups such as workaholics anonymous to recover from workaholism. These interventions aim to help workaholics overcome the negative consequences and regain a more sustainable and fulfilling approach to work, but also to support the organizations’ productivity that, subsequently, may be detrimentally impacted by workaholism consequences. Several studies calculated the economic burden of psychosocial risks and work-related stress: a systematic review [81] with data from Australia, Canada, Denmark, France, Sweden, Switzerland, the United Kingdom, and the EU-15 showed that the total estimated cost of work-related stress ranged from US $221.13 million to $187 billion. In addition, work-related stress contributes to around half of all lost working days [82].

### 5.2. Limitations and Further Research

The present study makes a contribution to the existing literature by examining the relationships between workaholism and technostress, shedding light on how the former may affect the latter. Nonetheless, it is important to consider the limitations of our study when interpreting the results. Firstly, the use of convenience sampling may introduce potential biases when interpreting our results and limit the generalizability of the findings. The sample size also does not allow a broad generalization of outcomes to the entire population of Italian workers. Secondly, this study relied on self-reported measures, which are susceptible to social desirability bias. This bias may be particularly relevant for workaholics who may feel pressured to present themselves in a favorable light, potentially affecting the accuracy of the data. Future research could consider using additional objective measures or alternative data collection methods to minimize this bias. Thirdly, as suggested by Burisch [83], the relatively short time lag adopted in this study was chosen to minimize attrition among participants and capture the perceptions of employees during a specific period, namely the pandemic, when the use of ICTs was intensified. Further studies could test the relationship between the focus variables over a longer lag of time. To consolidate our results, further research could delve into the processes underlying the workaholism–technostress relationship, considering, for instance, the mediating role of increased technology use or high job demands or their interplay with some job design features (e.g., amount or procedures of remote working). Additionally, examining the unique effects of different workaholism components and their interplay could provide a more nuanced understanding of the workaholism-technostress relationship. Future studies could also compare workaholics with non-workaholics, helping in understanding the differences in technostress experiences. Furthermore, it would be beneficial to investigate these relationships in the post-COVID-19 era, when the use of ICTs is likely to be more integrated into organizational practices and routines. This would provide insights into the long-term effects and implications of workaholism and technostress beyond the exceptional circumstances of the pandemic.

## 6. Conclusions

During the pandemic, the shift to remote work and other changes in work dynamics have brought attention to phenomena such as technostress and workaholism. While limited studies explored the relationship between these constructs, the existing evidence suggests an intertwined nature between workaholism and technostress. The current study aims to address this research gap by exploring the longitudinal relationship between workaholism and technostress. The results of the present study highlight that workaholism might serve as a potential antecedent to technostress. Employees with workaholic traits are more likely to use technology frequently because it allows them to extend their normal working hours and engage in additional work-related tasks during non-working hours. Despite these findings, there are still open questions that warrant further investigation in future studies.

One important aspect for further research is to verify whether these results persist beyond the COVID-19 era, analyzing whether and how the relationship between workaholism and technostress persists in different work contexts and under different circumstances. By addressing these open questions, future studies can provide a more comprehensive understanding of the dynamic interplay between workaholism and technostress, shedding light on the long-term implications and effects of these constructs on employees’ well-being and work performance.

Moreover, future studies could overcome the convenience sampling method and self-report limits by employing a more diverse and representative sample, as well as incorporating multiple data sources for a comprehensive analysis. Additionally, concerning the time lag used in this study, it would be interesting to examine the relationship between workaholism and technostress using longitudinal designs with multiple waves and follow-up assessments to examine the stability of the findings over time. In addition, according to studies that conceptualize workaholism as a state condition with variations within the individual [45], it would be interesting to examine the relationship using daily measurements to capture these variations.

A further avenue for future research could involve investigating the impact of work context and other variables that might affect the process from workaholism to technostress. For instance, future studies could investigate the moderating role of high job demands or organizational support, which represent organizational factors that can mitigate or exacerbate the negative effects of workaholism and the subsequent perception of technostress [25,42,43]. By considering these contextual factors, a more comprehensive understanding of the mechanisms underlying the transition from workaholism to technostress can be obtained. Understanding the specific conditions under which workaholism leads to technostress can inform the development of interventions and policies aimed at reducing technostress and promoting a healthier work–life balance.

Finally, examining the profiles of workaholic and non-workaholic workers can help differentiate the effects of workaholism on technostress and provide tailored interventions for individuals at risk. From a practical and professional point of view, our results may help managers and organizations in identifying and addressing promoting healthier work habits and better technology management. These findings can also help raise awareness among managers about the potential risks associated with workaholism and technostress. By increasing awareness of these phenomena and their detrimental effects, organizations can develop strategies and interventions to foster a more balanced and sustainable use of technology and promote employee well-being.

Overall, future research should continue to explore the complex interplay between workaholism, technostress, and other organizational and individual factors to inform evidence-based interventions and practices to create healthier work environments and improve employees’ experiences with technology.

## Figures and Tables

**Figure 1 behavsci-13-00599-f001:**
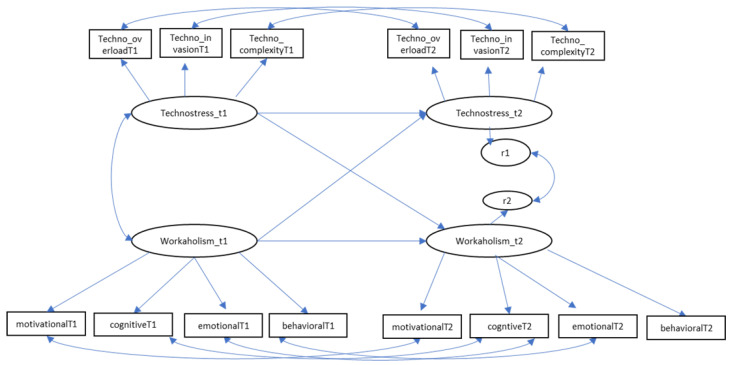
Model of cross-lagged associations between technostress and workaholism.

**Figure 2 behavsci-13-00599-f002:**
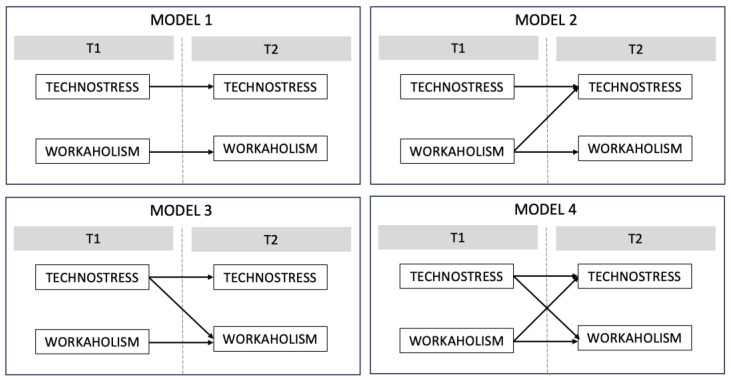
The tested models: Model 1: Autoregressive model; Model 2: cross-lagged pathway from workaholism at T1 to technostress at T2; Model 3: cross-lagged pathway from technostress at T1 to workaholism at T2; Model 4: cross-lagged associations between technostress and workaholism.

**Table 1 behavsci-13-00599-t001:** Descriptive statistics and correlation coefficients (in brackets) of studied variables.

Variables	M	S.D.	1	2	3	4	5	6
Workaholism (T1)	2.70	0.72	(0.88)					
2. Technostress (T1)	2.46	0.85	0.26 **	(0.89)				
3. Workaholism (T2)	2.57	0.89	0.77 **	0.28 **	(0.93)			
4. Technostress (T2)	3.01	0.94	0.33 **	0.77 **	0.42 **	(0.91)		
5. Gender	–	–	0.18	0.12	0.19 *	0.09	–	
6. Age (T1)	34.31	11.08	−0.03	0.11	−0.05	0.10	−0.05	–

Note: * *p* < 0.05, ** *p* < 0.001; T1 = Time 1 and T2 = Time 2; Gender was coded as 1 = men and 2 = women.

**Table 2 behavsci-13-00599-t002:** Model fit statistics for measurement invariance testing.

Models	χ^2^ (df)	CFI	TLI	RMSEA	SRMR
Workaholism (T1)	148,952 (98)	0.946	0.933	0.068	0.084
Workaholism (T2)	150,509 (98)	0.961	0.952	0.069	0.074
Technostress (T1)	59,262 (41)	0.977	0.969	0.063	0.064
Technostress (T2)	75,967 (41)	0.962	0.949	0.087	0.053

**Table 3 behavsci-13-00599-t003:** Model fit statistics for the cross-lagged path model.

Models	χ^2^ (df)	CFI	TLI	RMSEA	SRMR
Model 1 Autoregressive	96,286 (67)	0.966	0.954	0.062	0.069
Model 2 Cross-lagged	82,550 (65)	0.980	0.971	0.049	0.057
Model 3 Cross-lagged	85,132 (65)	0.977	0.967	0.053	0.059
Model 4 Reciprocal cross-lagged	81,092 (64)	0.980	0.972	0.049	0.056

**Table 4 behavsci-13-00599-t004:** Parameter estimates of the cross-lagged path model.

Paths	B	S.E.	C.R.
Model 1 Relationship between workaholism and technostress			
Technostress (T1) → Technostress (T2)	0.95 **	0.116	8.160
Workaholism (T1) → Workaholism (T2)	0.97 **	0.140	6.904
Workaholism (T1) → Technostress (T2)	0.25 *	0.126	1.966
Technostress (T1) → Workaholism (T2)	0.08	0.065	1.215

Note: ** *p* < 0.001 * *p* < 0.05; B = unstandardized coefficients; S.E. = standardized error of the estimate; C.R. critical ratio.

## Data Availability

The data presented in this study are not publicly available due to Italian privacy law. The data are available on request from the corresponding author.

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
