# Peer review of "The Workaholism–Technostress Interplay: Initial Evidence on Their Mutual Relationship"

_behavsci, 2023, doi:10.3390/bs13070599_

Round 1

Reviewer 1 Report

Thank you for the opportunity to read this very interesting paper. The paper has many strengths that should be of interest to the journal audience. Thus, the following suggestions are around enhancing the presentation for publication and clarifying aspects of the data and reporting.

I will go by line number for the most part. If not, I will try to be as specific as possible in noting the area I am speaking about.

Abstract

Purpose. It is necessary to include the country.

[Line 13-15] The objective should detail the variables and authors shouldn’t combine results data (113 employees). Where do the employees work?

[Line 14] Authors must specify the type of study design. A cross-sectional study was carried out with a sample…

[Line 15-17] Where are p-values? (p < .001). Authors must specify it. P values showing the differences between groups should be given.

Keywords. Technostress; Workaholism; cross-lagged panel study are not MeSH Terms. Also, I recommend to you include COVID-19, mental health, work conditions, among others.

1. Introduction

[Line 67-69] Why do you include here the main aim?

2. Materials and Methods

There aren’t enough details to repeat the experiments and it isn’t clear how samples were collected or how participants were recruited.

Did the authors calculate the needed sample size? Please, clarify. How was the sample size determined? Did the authors test power calculation? How was the sample chosen? It is also necessary to specify the inclusion and exclusion criteria for the study sample.
Authors must specify it.

Do the authors have a study protocol? The study protocol should be described in detail.

[Lines 180-188] This is information should be in Results section.

Are the scales adapted to the Italian population? Which are breakpoint? Authors must justify their response.

Ethical considerations. You should include a sub-heading under Methods that describes these issues: provide the reference number of the Ethics Committee approval, describe how the confidentiality of the data has been guaranteed. Please include the date and code register number of ethics committee.

Statistical analysis. It is convenient to run, and describe, the analysis of normality in the distribution of scores of the validated questionnaires they have used, in order to justify the use of parametric or non-parametric contrast statistics.

3. Results

Demographic data on respondents should be given.

The results are presented clearly and accurately.

At last, but not least, I recommend you to make available your data in an open repository. I think it will make this scientific process more transparent, and it allows other researchers to replicate your results.

4. Discussion

The authors logically explain the findings.

Limitations of the study weren’t discussed Limitations related to the type of methodology used. Limitations regarding the representativeness of respondents should be better addressed Authors must specify it. The fact of having a convenience sample should be included in the limitations of the study.

Please, provide the conclusions.

I wish you all the best.

Author Response

We thank  Reviewer for comments on our manuscript. Please see the attachment.

Sincerely,

On behalf of all the authors,

Carmela Buono

Reviewer 2 Report

The topic is interesting, according to the period we went-through, but in order to be published the authors must make some important improvements:

1. In the abstract, the results are presented in a general manner, they must refer to  them using some data.

2. In the Introduction the authors must add in the final part about the novelty of the study, and more important, each chapter developed in the article must be shortly presented.

3. In the section 3.1. the authors only remind us about some developed affirmations, so, in Table 1 the authors must add all the affirmations for each analysed variables (from 1 to 4), and calculate for them all the indicators reminded at section 3.3. (Cronbach’s Alpha, standard deviation, MD, skewness, kurtosis).

4. As we observe workaholism 1 and 2 are mediators, so is a must to analyse their role in the two states as mediators.

5. The relationships from Table 4 must be interpreted according to B, CR and SE (and their limit value added and analysed according to them).

6. There is only a short Discussion, without saying if the research hypothesis were or weren’t fulfilled.

7. The Conclusion and proposal section is missing, so is important to add them, especially to add the resuts obtained and their implications, the limitations, future research directions and the implications (for organizations from Discussion) must be developed, especially for employees, and for society.

8. More important, the authors must add at least 10-15 sources which are from 2022-2023. They must be up-dated, because the sources are old, only two are from 2022...

Thus, in this moment the authors must improve the content of the study (as 1-8 issues), and are required major revisions in order to be proposed for publishing.

Author Response

(The authors gave the same response as above.)

Round 2

Reviewer 1 Report

All my recommendations were implemented.

Please, review Covid-19 by COVID-19 in Keywords.

The latest version of the Helsinki Declaration (Fortress Amendment, Brazil, October 2013).

Author Response

(The authors gave the same response as above.)

Reviewer 2 Report

A few, but important suggestions were improved, but the following suggestions suffered to be improved (some were partially improved or not at all).

4.As we observe workaholism 1 and 2 are mediators, so is a must to analyse their role in the two states as mediators.

6.In Discussion, the authors must add if the research hypothesis were or weren’t fulfilled, each one.

7.The Conclusion is very short, it must be improved, and the implications remained the same, only for organizations, so, they must be developed, especially for employees, and for society.

Thus, in this moment the authors must improve the content of the study (4, 6 and 7 issues), and are required major revisions in order to be proposed for publishing.

Author Response

(The authors gave the same response as above.)

Round 3

Reviewer 2 Report

As was mentioned, the Conclusions are too short, and must be improved by:

-        Offering the answer to the research question,

-        Synthesizing ideas and represents an opportunity to extend research findings. Extensive documentation will lead to the writing of relevant conclusions (Giltrow et al., 2021).

-        Beginning with the presentation of the authors own points of view regarding the studied topic; the authors must present the limits of the research, the implications of the results, offering an opening to other research.

Author Response

(The authors gave the same response as above.)
